# Generating Global Factual and Counterfactual Explainer for Molecule under Domain Constraints

**Danqing Wang** [* 1]  **Antonis Antoniades** [* 1]  **Ambuj Singh** [1]  **Lei Li** [1]

## Abstract

Graph neural networks (GNNs) are powerful tools for handling graph-structured data but often lack transparency. This paper aims to generate interpretable global explanations for GNN predictions, focusing on real-world scenarios like chemical molecules. We develop an approach that produces both factual and counterfactual explanations while incorporating domain constraints, ensuring validity and interpretability for domain experts. Our contributions include creating global explanations, integrating domain constraints, and improving random walk in global explanations using fragment-based editing. We demonstrate the effectiveness of our approach on AIDS and Mutagenicity datasets, providing a comprehensive understanding of GNNs and aiding domain experts in evaluating generated explanations.

## 1. Introduction

Graph neural networks (GNNs) are a natural and beneficial choice when the underlying data distribution can be characterized by a graph structure, such as social network (Yanardag & Vishwanathan, 2015), chip design (Mirhoseini et al., 2021), chemical molecules (Gilmer et al., 2017; Wieder et al., 2020). However, the lack of transparency in GNNs preventing the full exploitation of the rich information available in both the topology structure and the node/edge features (Zhang et al., 2020; Zhou et al., 2020).

As a result, there is a growing demand for interpretable explanations of GNN predictions. Existing methods can be classified into local and global explanations based on their granularity. Local explanations generate explanations for individual instances by selecting the most influential sub-graph for that particular prediction (Bajaj et al., 2021; Lucic et al., 2022; Tan et al., 2022), while global methods create a set of explanations for a specific class to explain why GNNs predict them as that class (Ying et al., 2019; Kosan et al., 2023). Furthermore, these methods can be categorized as factual or counterfactual explanation methods. Counterfactual explanations identify necessary sub-graphs that can alter model predictions (Lucic et al., 2022; Bajaj et al., 2021; Abrate & Bonchi, 2021), while factual explanations focus on sufficient parts that maintain the original explanations (Ying et al., 2019; Luo et al., 2020; Yuan et al., 2020).

The local explanations are known for the vulnerability to the small noise in the input graph (Bajaj et al., 2021) and the lack of high-level insight to explain the behavior of GNNs (Kosan et al., 2023). Therefore, our work focuses on generating global interpretable explanations for GNNs, particularly in real-world scenarios like chemical molecules. Previous global methods concentrated on either factual or counterfactual aspects, insufficient for a comprehensive understanding of GNNs (Tan et al., 2022). Although some research overcomes the limitation of sub-graph-as-explanation and uses common-neighbor-graph-as-explanation for molecules, it still disregards the validity of new molecule candidates, complicating explanations for domain experts (Kosan et al., 2023).

In this paper, we develop both factual and counterfactual explanations for molecules to examine the sufficiency and necessity conditions of undesirable chemical attributes from a GNN classifier viewpoint. To enhance domain experts' understanding, we incorporate domain constraints in our explainer, ensuring the validity of molecules that can be easily evaluated and verified by experts.

To conclude, our contributions can be described as:

- We develop both factual and counterfactual explanations using a global perspective, examining graph space and embedding space.

- We explore various approaches to integrate domain constraints into global explanations, enhancing interpretability for domain experts.

- We advance fragment-based editing to refine the random walk in global explanations, increasing their effectiveness in the Mutagenicity dataset.

---

[*]Equal contribution  [1]UC Santa Barbara, California, USA. Correspondence to: Danqing Wang <danqingwang@ucsb.edu>.

*Workshop on Interpretable ML in Healthcare at International Conference on Machine Learning (ICML)*, Honolulu, Hawaii, USA. 2023. Copyright 2023 by the author(s).

## 2. Related Work

**Explanations of Graph Neural Networks**   Prior studies aimed at explaining GNNs can be divided into two categories: local and global explainers. Local explainers select sub-structures from a given graph that contribute to its GNN's prediction (Ying et al., 2019; Lucic et al., 2022; Tan et al., 2022), whereas global explainers produce new graphs to illustrate the model behavior across a set of graphs (Kosan et al., 2023; Yuan et al., 2020). The generated explanations pivot on either factual or counterfactual reasoning. Factual reasoning yields an explanation preserving the original prediction, thus acting as a sufficient condition. Conversely, counterfactual reasoning presents a necessary condition that would, if not met, alter the prediction (Tan et al., 2022). Initially, research predominantly focused on factual explanations (Ying et al., 2019; Luo et al., 2020; Yuan et al., 2020), however, recent trends indicate growing interest in counterfactual explanations (Bajaj et al., 2021; Lucic et al., 2022; Kosan et al., 2023; Numeroso & Bacciu, 2021). Tan et al. (2022) discuss the advantages and disadvantages of and combine them together. Yet, no previous work has explored the application of global factual and counterfactual explanations.

**Evaluation of GNN Explanations**   It is difficult to create GNN datasets with annotated explanations. Previous studies often evaluate their methods on the small synthetic datasets and conduct human evaluation on limited cases of real datasets (Ying et al., 2019; Yuan et al., 2020; Luo et al., 2020). Given the ground-truth labels, they calculate the accuracy of prediction as the evaluation metric. Pope et al. (2019) introduced fidelity to measure the decrease of prediction confidence after removing the explanation for counterfactual explanations. Bajaj et al. (2021) proposed robustness for quantifying how much an explanation changes after adding noise to the input graph. Tan et al. (2022) adopted Probability of Sufficiency (PS) and Probability of Necessity (PN) from causal inference theory for factual and counterfactual explanations. Kosan et al. (2023) propose coverage and cost for global explanations. Amara et al. (2022) extended fidelity to model-level for both sufficiency and necessity explanations. In this paper, since we do not have the ground-truth labels and focus on global level, we follow Kosan et al. (2023)'s setting to use coverage and cost as the evaluation metrics.

**Graph-based Molecule Generation**   Graph neural network are widely used in 2D molecule tasks (Gilmer et al., 2017; Wieder et al., 2020).A molecule can be graphically represented with atoms as vertices and chemical bonds as edges. The atom-based generation methods take the atom as the basic generation units (Li et al., 2018; You et al., 2018), while the fragment-based methods build their vocabulary based on the chemical substructure (Jin et al., 2018; Liu et al., 2017; Kong et al., 2022). The fragment-based generation is more likely to produce meaningful molecule with chemical properties, which is reflected in the substructure. Besides, it can make the edit-based sampling more effective and efficient (Xie et al., 2021). In this paper, we use the fragment-based editing to ensure the validation of the molecule candidates.

## 3. Method

Given a graph classifier $\phi$ and a set of $n$ input molecule $\mathcal{G} = \{G_1, G_2, \cdots, G_n\}$, we assume $\phi(G_i) = 0$ indicates that $G_i$ has the undesirable attribute, while $\phi(G_i) = 1$ represent that it has the desired attribute. Following Kosan et al. (2023), The goal of the global explanation is find a small set of valid molecules $\mathcal{C}$ with optimal explanation ability. The explanation ability is evaluated by **coverage**, **cost** and **size**. The size is the number of graphs in the set $\mathcal{C}$. The coverage is to measure how many input graphs $G \in \mathcal{G}$ can be covered by explanations in $\mathcal{C}$ under a given distance threshold $\theta$:

$$\textbf{coverage}(\mathcal{C}) = \frac{|\{G \in \mathcal{G} | \min_{C \in \mathcal{C}} d(G,C) \le \theta\}|}{|\mathcal{G}|}, \quad (1)$$

where $d(G,C)$ is the function to calculate graph distance, and $|\mathcal{G}|$ indicates the size of the set $\mathcal{G}$. The cost is the distance between the input graphs $\mathcal{G}$ and the explanations $\mathcal{C}$:

$$\textbf{cost}(\mathcal{C}) = \frac{1}{|\mathcal{G}|} \sum_{i=1}^{|\mathcal{G}|} \min_{C \in \mathcal{C}} d(G,C). \quad (2)$$

We would like to maximize the coverage while minimize and cost and size to make the explanation set $\mathcal{C}$ cover as many input graphs as possible while keeps small enough for human cognition. Therefore, the objective is :

$$\max_{\mathcal{C}} \textbf{coverage}(\mathcal{C}) \quad s.t. \textbf{size}(\mathcal{C}) = k. \quad (3)$$

Here, the size of the $\mathcal{C}$ is limited by k, and the cost is constrained based on the threshold $\theta$ in coverage.

To obtain global explanations, we generate candidates from the input molecule. We edit the input molecule by adding, deleting, or replacing nodes and edges, creating a meta edit map $E_\mathcal{G}$. Each node $v \in E_\mathcal{G}$ in the meta edit map represents a modified molecule derived from the original input molecules through the defined edit operations. The random walk is based on the importance score calculated on each node $v$:

$$I(v) = p(\phi(v) = 1)(\alpha \, \textbf{coverage}(v) + (1 - \alpha) \, \textbf{gain}(v)), \quad (4)$$
$$\textbf{gain}(v) = \textbf{coverage}(\mathcal{C} \cup \{v\}) - \textbf{coverage}(\mathcal{C}) \quad (5)$$

### 3.1. Exploring Counterfactual Explanations under Domain Constraints

Random editing operations based on atoms, such as adding, removing, or replacing nodes and edges, often produce numerous invalid molecules that are meaningless to domain

experts. To make explanations more comprehensible, we must integrate domain constraints, like valence, into the random walk process.

One straightforward approach involves verifying validation after candidate generation and discarding illegal candidates. We refer to this method as **Post-checking**. Alternatively, we can modify the importance function to ensure that it only moves to valid nodes during the random walk. The updated importance function is as follows:

$$I(v) = \begin{cases} p(\phi(v))(\alpha \, \textbf{coverage}(v) + (1 - \alpha) \, \textbf{gain}(v)) & f(v) = 1 \\ 0 & f(v) = 0 \end{cases}$$

(6)

Here, $f(v)$ is the verification function and $f(v) = 1$ indicate $v$ is a valid molecule. This method is denoted by **In-checking**.

### 3.2. Fragment-based editing

Throughout our experiments, we found that only a limited number of editing operations yield new valid molecules, while most result in valence violations. To enhance search efficiency, we transition from atom-based editing to fragment-based editing, which exclusively adds or deletes legal fragments.

To construct a fragment vocabulary, we follow Xie et al. (2021) by breaking each single bond of molecules in the dataset and considering the smaller arms as separate fragments. During the random walk, we implement fragment addition and removal for the current node. For addition, we enumerate node positions to insert the fragment and test all fragments in the vocabulary. For removal, we break single bonds and delete the smaller fragment. After each edit, we verify their validation. This process is depicted in Figure 1.

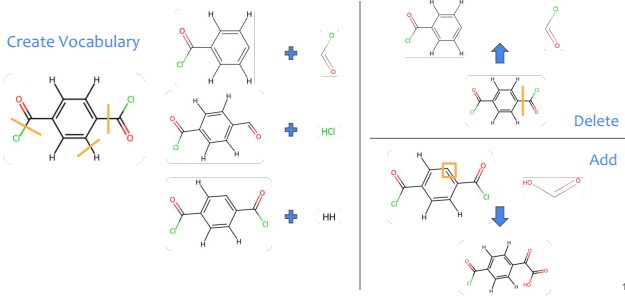

*Figure 1.* Fragment-based Editing.

### 3.3. Global Factual and Counterfactual Explanations

To better comprehend the GNN classifier's behavior, we generate factual and counterfactual explanations near the decision boundary, serving as the classifier's supporting points. By comparing candidates on both sides, we can identify significant patterns for different categories and determine the distance between them.

As described in Section 3, we initially create global counterfactual explanations for graphs with undesirable attributes ($\phi(G) = 0$). As counterfactual explanations provide necessary conditions for altering predictions, candidates will have $\phi(G) = 1$ (desirable attributes). These counterfactual explanations offer insights on transforming undesirable attributes into desirable ones.

To acquire factual explanations for undesirable attributes, we use the same input ($\phi(G) = 0$) and change the counterfactual probability $p(\phi(v) = 1)$ to factual probability $p(\phi(v) = 0)$ in the importance function:

$$I(v) = p(\phi(v) = 0)(\alpha \, \textbf{coverage}(v) + (1 - \alpha) \, \textbf{gain}(v)). \quad (7)$$

However, since the original input graphs $G$ have label 0, factual explanations might directly use the input graphs, leading to oversimplified explanations far from the decision boundary.

To generate factual explanations closer to the decision boundary, we employ another counterfactual explanation generation for input molecules with desirable attributes ($\phi(G) = 1$). This process generates candidates with label=0, but factual and counterfactual explanations are not for the same input graphs. Our goal is to generate global explanations for the GNN classifier to understand why it predicts a specific label, rather than instance-level explanations for each input. Hence, it is unnecessary to use the same input graphs to create explanations.

## 4. Experiments

### 4.1. Datasets

We focus on graph classification and conduct our experiments on two real-world molecule datasets that are commonly used in graph classification: AIDS (Riesen et al., 2008) and Mutagenicity (Kazius et al., 2005). Detailed information is listed in Table 1. These datasets provide essential node information (atom type) and edge attributes (bond type) for molecules. Following Kosan et al. (2023), we keep atom types that appear at least 50 times in the dataset, resulting in 9 common atoms in AIDS and 10 in Mutagenicity.

### 4.2. Implementation

Our method is implemented using Pytorch. We employ the *rdkit* library for molecular operations. First, we convert the original graph into the molecule format by re-mapping the one-hot node feature to atom type and using the edge attribute as bond type. We then use *rdkit.Chem.detectChemistryProblems* to check validity, which identifies and captures error messages when creating molecule objects. We evaluate performance using coverage

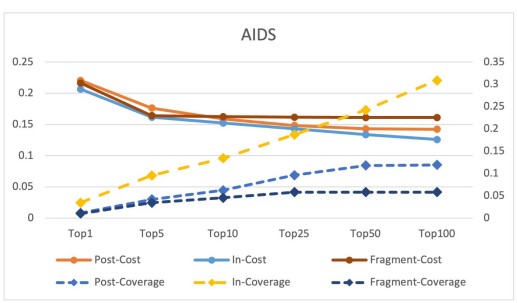 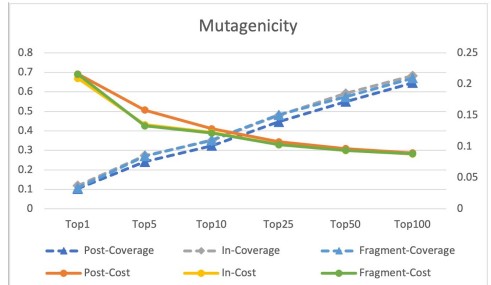

*Figure 2.* Coverage and cost for different candidate size.

*Table 1.* Dataset Statistic. Here # indicates the number size. We use the GNN classifiers trained by Kosan et al. (2023) to predict the category.

|  | AIDS | Mutagenicity |
|---|---|---|
| #Graphs | 1837 | 4308 |
| #Nodes per graph | 15.73 | 30.34 |
| #Edges per graph | 16.32 | 30.80 |
| #Atom Type | 9 | 10 |
| #Graph label=0 (pred/true) | 1473/1467 | 2438/2394 |

and cost as defined in Section 3. For fragment-based editing, we first build the vocabulary for fragments following Xie et al. (2021) for each dataset, and enumerate all potential edit operations, which is similar as the atom-based editing. And then we traverse the meta edit map $E_\mathcal{G}$ based on the importance function in Equation 6.

### 4.3. Explanations under Domain Constraint

We compare different methods for incorporating domain constraints into global explanation generation in Table 2. We see that compared to the original GCFExplainer, the performance degrades in the Mutagenicity dataset but improves in AIDS under the in-checking settings. We assume this is because the graphs in AIDS are relatively simple but exhibit various patterns. Adding validity checking actually prunes the illegal ones and prevents the random walk from reaching some local optima. However, for fragment-based methods, the patterns between different input graphs are so diverse that they can hardly be used for other graphs. Simultaneously, fragment-based editing cannot apply simple modifications based on atoms, which limits exploration in AIDS. Conversely, although the Mutagenicity dataset is much larger than AIDS, the fragment vocabulary size is small. It suggests that the graphs share similar patterns. Thus, a fragment from one graph is likely to be helpful for another graph. Therefore, the fragment-based method is more efficient than the other two checking methods.

We also compare the coverage and cost between different k candidates. As illustrated in Figure 2, we find that the

performance gap in AIDS increases as the size of candidates increases. This indicates that the random walk in AIDS heavily depends on the importance function score and can easily be trapped in local optima.

### 4.4. Comparing Factual and Counterfactual Explanations

**Case Study** As shown in Figure 3, factual explanations are usually larger and can cover more input graphs. However, they are sufficient but not necessary. For example, in Figure 3 (f) the explanation only covers one input graph, but it is much larger than the original one. Conversely, counterfactual explanations typically cover 1 or 2 graphs. Nonetheless, there are evident patterns that can alter the classifier's behavior, such as HCl to $PH_3$, and H to HF.

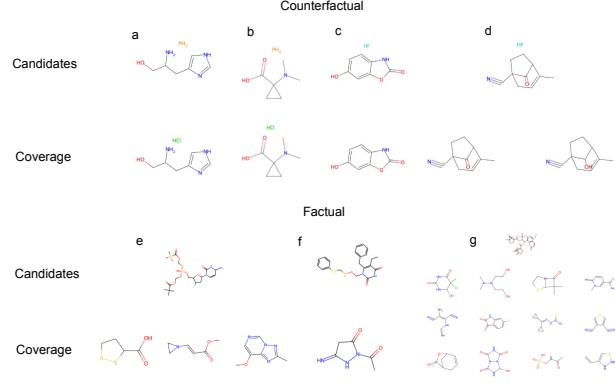

*Figure 3.* Counterfactual and Factual Candidates in AIDS.

**Embeding Space** In the process of generating factual and counterfactual explanations, we navigate a highly non-smooth, high-dimensional decision boundary from both sides for the respective explanations. Furthermore, comparing these two types of explanations can help narrow down the space of potential, useful attributes. Although we lack the expertise to manually evaluate the explanations, we attempt to analyze them at the feature level.

*Table 2.* Comparison of different methods. Time is calculated based on minutes.

| | AIDS (vocab=1000) | | | | Mutagenicity (vocab=408) | | | |
|---|---|---|---|---|---|---|---|---|
| | Coverage ↑ | Cost↓ | #Valid / #Total | Time (') | Coverage↑ | Cost↓ | #Valid/#Total | Time (') |
| GCFExplainer | 12.56% | 15.35 | / | / | 38.52% | 11.37 | / | / |
| Post-checking | 6.25% | 15.89 | 114/868 | 9 | 32.36% | 12.84 | 883/2438 | 258 |
| In-checking | **13.44%** | **15.25** | 1473/1473 | 15 | 34.95% | 12.23 | 2438/2438 | 242 |
| Fragment-based | 4.55% | 16.25 | 125/125 | 365 | **35.32%** | **12.15** | 2438/2438 | 229 |

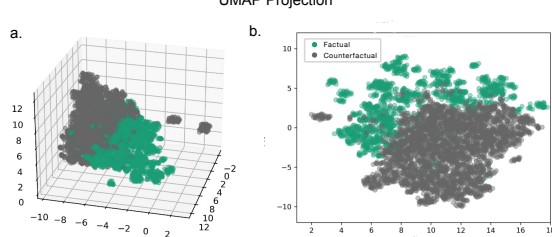

*Figure 4.* UMAP projections of the features in 2D (a) and 3D (b) in AIDS.

We first convert the graphs into embeddings and then project a UMAP representation of them. In Figure 4, we observe that counterfactual and factual explanations are somewhat separable, with significant regions of overlap along a tight boundary. This could indicate the difficulty of explaining transitions and classifying molecules.

**Graph Space** Having examined the relationship between explanations in the embedding space, we also want to explore it in the graph space. We choose the candidates with the most coverage from the respective generated sets and compare the pairwise distance between them. As shown in Table 3, we find that there is a substantial distance between the graphs, especially for comparisons involving larger sets. Interestingly, we also discover some graphs with very close distances to each other.

*Table 3.* Graph Edit Distance (GED) between factual and counterfactual explanations.

| Candidate | Top1 | | Top5 | | Top10 | | Top-25 | |
|---|---|---|---|---|---|---|---|---|
| | | | Min | Avg. | Min | Avg. | Min | Avg. |
| AIDS | 43.0 | | 28.0 | 60.2 | 27.0 | 90.4 | 13.0 | 95.8 |
| Mutagenicity | 13.0 | | 4.0 | 23.0 | 4.0 | 23.5 | 4.0 | 30.9 |

## 5. Conclusion and Discussion

In this paper, we examine global factual and counterfactual explanations for GNN classifiers. We find common patterns in these explanations that help us better understand the model's behavior. Moreover, although these explanations differ significantly in graph space, it remains challenging to identify a clear decision boundary in the embedding space. To incorporate domain constraints, we explore various methods and discover that in-checking provides a better guideline for datasets with diverse patterns, while fragment-based methods are effective when the vocabulary is representative.

We believe that these techniques could have a substantial impact in areas such as medicine and drug discovery. To advance in this direction, there are some obvious next steps: 1) Further investigate the relationship between global factual and counterfactual explanations. 2) Continue refining validity checking and fragment-based editing. 3) Convert graph networks to 3D.

Additionally, more ambitious goals could be pursued. The explanations are limited by the quality of the sub-graph classifier, which is not optimal. An interesting avenue to explore would be to enhance our system by using the explanations to improve the classifier and vice versa, creating a self-reinforcing loop. Finally, involving humans in this loop to provide feedback on the desirability and validity of the explanations could further enhance the system.

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
