# OpenReview forum: "Generating Global Factual and Counterfactual Explainer for Molecule under Domain Constraints"
_ICML.cc/2023/Workshop/IMLH — IMLH 2023 PosterShortPaper_

### Official Review · Reviewer_spuh · 2023-06-16
**I am not an expert in this area.**

**Rating:** 4
**Confidence:** 2

**Review:**

+ The paper is written with good logic.
+ The experiments are well-designed.

- The method seems to lack novelty.
- The experimental results are not very strong.

---

### Official Review · Reviewer_YQnZ · 2023-06-17
**Global explanation of GNN**

**Rating:** 7
**Confidence:** 3

**Review:**

The paper presents a compelling approach to address the transparency issue in graph neural networks (GNNs) when dealing with graph-structured data, with a specific focus on chemical molecules. The authors' development of global interpretable explanations for GNN predictions is a significant contribution to the field. By incorporating both factual and counterfactual explanations, the authors provide a comprehensive understanding of GNN behavior. The integration of domain constraints ensures the validity of generated molecules, enabling domain experts to evaluate and verify the explanations effectively. The advancement of fragment-based editing techniques improves the random walk in global explanations, enhancing their effectiveness, particularly in the Mutagenicity dataset. The experimental results on the AIDS and Mutagenicity datasets showcase the efficacy of the proposed approach in providing interpretable explanations for GNN predictions. Overall, this paper is a valuable contribution to the field of interpretable GNNs, addressing real-world scenarios and aiding domain experts in evaluating the generated explanations.

Personally, I enjoyed reading the manuscript, it very clearly written. While the experiments are sufficient for this workshop, i have following points which will make the paper stronger:
1. The authors aim to help the domain experts. So, in future they should conduct a user study proving that their method is better than the baselines.
2. Lack of baseline comparison. The authors only compare the efficacy of the method with GCFExplainer. They should add more baselines.
3. While the authors showed the qualitative evaluation of the interpretability aspect, they should use some interpretability metric to quantify.
They should follow the following 2 papers on how to evaluate / create interpretability metric:

    [1] GraphFramEx: Towards Systematic Evaluation of Explainability Methods for Graph Neural Networks, Amara et al.

    [2] Encoding Concepts in Graph Neural Networks, Magister et al.

4. [Nice to have] Can the authors add concept based explainability like [2], because extracting concepts from GNN is really an important tool to explain the prediction.

Overall it is a good read, and best of luck for the futue.

---

### Official Review · Reviewer_tkgE · 2023-06-18
**Generates global counterfactual reasoning for graph classifiers. Introduces  domain constraint of checking chemical validity to improve random walk process. Short Paper**

**Rating:** 6
**Confidence:** 4

**Review:**

This paper attempts to provide insights of producing factual and counterfactual explainers for Graph classifiers. Importantly, instead of checking for validity (whether the produced molecule is chemically valid or not) as a post-hoc step authors incorporate it in the randomw walk stage itself. Moreover, they also propose to edit fragments rather than atoms to improve the search efficiency and decrease the number of chemically invalid molecules. Counterfactual generation optimization is often not well-regularized and having these domain constraints can provide a means to better regularize. While the paper focuses on the global explanations as opposed to local, it would have been better if they showed the performance with local counterfactual generation methods. As it stands, imo the current evaluation is not comprehensive. However, the UMAP analysis looks interesting and I appreciate the honesty of the authors in acknowledging the lack of expertise. In terms of paper writing, I would encourage authors to include a related work section and to give more details on how they generate the fragments.

---

### Official Review · Reviewer_3Zdw · 2023-06-18
**Review from Reviewer 3Zdw**

**Rating:** 5
**Confidence:** 3

**Review:**

Generating explanations for GNNs is quite relevant to the topic of IMLH. The paper focus on generating a global-level explainer for molecule, with necessary validation-based constraints.

**Strengths**
- Great visualization about molecule editing and case studies.

**Weaknesses**
- Authors squeeze a lot of technical details into 4 pages, which imposes challenges to fully understanding the proposed technique.
    - There is a logic gap in Sec2. Method. "The goal is to find a small set of valid molecules $C$ that can explain the global behaviors of $\phi$". It would be better to express under which measurement such $C$ can explain the global behaviors. I would suggest using a formula in this case.
    -  Why "optimize $C$ towards coverage and cost"? It would be better to add rationales for this argument.
    - Without referring to the GCFExplainer paper, there is no definition of what is $(\phi(G)=0)$. I suggest revising Sec2.3 to make it self-explainable.

---

### Meta-Review · Area_Chair_zRYK · 2023-06-19

**Recommendation:** Accept (Poster)
**Confidence:** 3

**Metareview:**

This short paper proposes a global XAI method with domain constraints on GNN. The method considers the domain specifications and is relevant in solving real-world problems. The authors should consider incorporating the reviewers comments in the revised version.

---

### Decision · Program_Chairs · 2023-06-20

Accept (Poster Short Paper)